# Explicit equivalence between the spectral localizer and local Chern and winding markers

Lucien Jezequel[1]*, Jens H. Bardarson[1], and Adolfo G. Grushin[2,3,4]

**1** Department of Physics, KTH Royal Institute of Technology, Stockholm 106 91, Sweden
**2** Donostia International Physics Center (DIPC), Paseo Manuel de Lardizábal 4, 20018, Donostia-San Sebastián, Spain
**3** IKERBASQUE, Basque Foundation for Science, Maria Diaz de Haro 3, 48013 Bilbao, Spain
**4** Univ. Grenoble Alpes, CNRS, Grenoble INP, Institut Néel, 38000 Grenoble, France

*lucienj@kth.se

October 9, 2025

## Abstract

**Topological band insulators are classified using momentum-space topological invariants, such as Chern or winding numbers, when they feature translational symmetry. The lack of translation symmetry in disordered, quasicrystalline, or amorphous topological systems has motivated alternative, real-space definitions of topological invariants, including the local Chern marker and the spectral localizer invariant. However, the equivalence between these invariants is so far implicit. Here, we explicitly demonstrate their equivalence from a systematic perturbative expansion in powers of the spectral localizer's parameter $\kappa$. By leveraging only the Clifford algebra of the spectral localizer, we prove that Chern and winding markers emerge as leading-order terms in the expansion. It bypasses abstract topological machinery, offering a simple approach accessible to a broader physics audience.**

# 1 Introduction

Topological band insulators display a bulk band gap populated by gapless boundary states [1]. The number of gapless boundary states is a topological invariant, remaining unchanged as long as the bulk remains gapped or features a mobility gap. The bulk-boundary correspondence [2–5] directly connects the number of boundary modes with a topological index defined in the bulk [6].

It is common to use translational invariance to express topological indices as integrals over the Brillouin zone of a function of the momentum-space wave function and its derivatives. However, systems that lack translational symmetry, for example disordered [7–21] and even amorphous systems [22–51], or quasicrystals [52–56, 56–67] can also display topological properties. To characterize topology in these systems, it is necessary to express topological indices as an integral over a local quantity, known as a local marker, in real space rather than in momentum space.

Within all the topological classes it is the class of two-dimensional systems displaying a quantized Hall effect [68, 69] that can be identified with the largest variety of local topological markers. o calculate the topological invariant for this class, the Chern number, in real-space, we can choose to Fourier transform the momentum-space formula to arrive to the so-called local Chern marker formula [9, 70–82]. The alternatives, the spectral localizer index [83–96] and the Bott index [84], rely on quantifying how close the position and Hamiltonian operators are to commuting, and on defining a trivial system as one where these operators commute. An earlier invariant, defined by Kitaev [76], is similar to the local Chern maker but based on a spatial tripartition. Additionally, it is possible to calculate the topological index using the rank difference of projectors [97]. Lastly, one can define a scattering invariant, which can be shown to be equivalent to counting edge modes [98].

The existence of this variety of markers for two-dimensional Chern insulator systems also poses the challenge of demonstrating their equivalence. It is possible to show that the Kitaev invariant, the Bott Index, the scattering invariant, and the local Chern makers are all equivalent [99, 100] [1]. However, an explicit connection between the localizer index and the rest is still obscure. Using algebraic topology, including K-theory [86, 101] and spectral flow analysis [85, 102], it is possible to show that the localizer index must be implicitly related to the local Chern marker. Ideally, it would be desirable to find an explicit derivation that relies on the simplest possible mathematics and connects to physical intuition. Finding such a derivation can be advantageous to propose other local markers for other topological classes. So far, only the spectral localizer and the scattering invariants apply all topological classes. Moreover, explicitly connecting the local Chern maker to the localizer index can help to better understand the regime of validity of the localizer index. Specifically, the spectral localizer requires choosing the magnitude of a scalar parameter $\kappa$ that determines the relative importance of the position and Hamiltonian operators within the spectral localizer. The criteria to choose the value of $\kappa$ is not well understood. It is usually chosen with guidance from rigorous mathematical bounds [86, 103], yet numerical simulations deliver correct results even when $\kappa$ violates these bounds [67, 104, 105]. Numerical evidence also shows that the phase diagrams computed for a Chern insulator quasicrystal using both the localizer index and the local Chern marker can be made to coincide for small values of $\kappa$ [67]. This suggest that a perturbation theory on $\kappa$ could be used to show the equivalence between these two methods.

In this work, we reveal how the equivalence between the localizer index and other local markers arises naturally from a perturbative expansion of the spectral localizer in powers of $\kappa$. By exploiting only the fundamental symmetries of the Clifford matrices used by the spectral

---

[1] For example, the Kitaev invariant is an integer exponentially quantized far away from edges [76] which reduces to the local Chern marker with periodic boundary conditions [99]. We thank Peru d'Ornellas for this comment.

localizer, we demonstrate that the Chern and winding markers emerge as the leading-order contribution in this expansion. Our derivation applies to the $\mathbb{Z}$ invariants in classes A and AIII of the Altland-Zirnbauer ten-fold classification of topological phases [2, 106]. Our derivation both circumvents heavy topological machinery, and establishes a direct, intuitive connection between the localizer index and the local Chern marker.

## 2  Brief introduction to local markers

### 2.1  Chern and winding markers

Our goal is to derive the integer topological invariants of the two complex classifying classes A and AIII of the Altland-Zirnbauer classification from the spectral localizer index. Depending on whether the system is defined in odd or even space dimensions, the topological invariants are defined as a winding or a Chern number, respectively [107]. In odd dimensions, only AIII hosts topological insulators from the two complex classes. This class has a chiral symmetry, represented by an operator $\hat{C}$ that anti-commutes with the Hamiltonian $[\hat{C}, \hat{H}]_{+} = \hat{C}\hat{H} + \hat{H}\hat{C} = 0$. In even dimension, only A hosts topological insulators from the two complex classes, and has no symmetry constraints.

In dimension $d$, the Chern ($C_{d/2}$) and winding ($W_{\lceil d/2 \rceil}$) numbers, can be defined in real space by a trace of the local Chern and winding markers, respectively. For systems with translational invariance the markers are defined as [70–72, 74, 75, 81]

$$d \textbf{ even}: \quad C_{d/2} = \quad \frac{-(-2i\pi)^{d/2}}{\Gamma(d/2+1)} \sum_{i_1,\ldots,i_d} \epsilon_{\vec{i}} \mathrm{Tr}_{x_0}\left( \hat{P} \prod_{k=1}^{d} [\hat{P}, \hat{x}_{i_k}] \right), \tag{1a}$$

$$d \textbf{ odd}: \quad W_{\lceil d/2 \rceil} = \quad \frac{-2(-2i)^{\lfloor d/2 \rfloor}\pi^{d/2}}{\Gamma(d/2+1)} \sum_{i_1,\ldots,i_d} \epsilon_{\vec{i}} \mathrm{Tr}_{x_0}\left( \hat{C}\hat{P} \prod_{k=1}^{d} [\hat{P}, \hat{x}_{i_k}] \right), \tag{1b}$$

where $\lceil \ \rceil$ represents the ceiling function, $\lfloor \ \rfloor$ the floor function, $\epsilon_{\vec{i}} = \epsilon_{i_1,\ldots,i_d}$ with $i_j \in [1, d]$ is the antisymmetric Levi-Civita symbol that equals $+1$ or $-1$ when the indices $i_1, \ldots, i_d$ form even or odd permutations of $1, 2, \ldots, d$, respectively, and equals 0 if any indices repeat. Additionally, $\Gamma(x)$ represents the Gamma function, $\hat{P}$ is the projector onto the occupied (negative-energy) states of the Hamiltonian $\hat{H}$, $\hat{x}_i$ are diagonal operators in position with diagonal elements $x_i$, and $\mathrm{Tr}_{x_0}$ denotes the trace over the internal degrees of freedom at a single position $x_0$.

In disordered systems, a single-site trace is insufficient to ensure quantization, and it is necessary to average spatially. To do so, we carry out the spatial trace by introducing a weight function $w(x)$ which averages over bulk sites and vanishes close to the boundary, with the normalization property $\sum_x w(x) = 1$ such that

$$d \textbf{ even}: \quad C_{d/2} = \quad \frac{(-2i\pi)^{d/2}}{2^{d+1}\Gamma(d/2+1)} \sum_{i_1,i_d} \epsilon_{\vec{i}} \mathrm{Tr}\left( \hat{w}\hat{H}_F \prod_{k=1}^{d} [\hat{H}_F, \hat{x}_{i_k}] \right), \tag{2a}$$

$$d \textbf{ odd}: \quad W_{\lceil d/2 \rceil} = \quad \frac{-(-2i)^{\lfloor d/2 \rfloor}\pi^{d/2}}{2^{d}\Gamma(d/2+1)} \sum_{i_1,i_d} \epsilon_{\vec{i}} \mathrm{Tr}\left( \hat{w}\hat{C}\hat{H}_F \prod_{k=1}^{d} [\hat{H}_F, \hat{x}_{i_k}] \right), \tag{2b}$$

where $\hat{w}$ is a diagonal operator in real space with diagonal elements $w(x)$. For later convenience, we have written Eqs. (2) in terms of the flattened Hamiltonian $\hat{H}_F = \mathbb{1} - 2\hat{P}$ instead of the real-space projector $\hat{P}$.

## 2.2  Spectral localizer and spectral localizer index

The topological invariants in all topological classes can be obtained from the spectrum of the spectral localizer operator [83, 103]. The spectral localizer operator depends on the Hamiltonian $\hat{H}$ and the position operator $\hat{x}$ as follows

$$d \text{ even}: \quad \hat{L} = \hat{H} \otimes \hat{\sigma}_{d+1} + \kappa \sum_{k=1}^{d} (\hat{x}_k - x_k') \otimes \hat{\sigma}_k, \tag{3a}$$

$$d \text{ odd}: \quad \hat{L} = \hat{H} \otimes \mathbb{1} + \kappa \sum_{k=1}^{d} (\hat{x}_k - x_k') \hat{C} \otimes \hat{\sigma}_k. \tag{3b}$$

It is defined using auxiliary degrees of freedom represented by the matrices $\sigma_i$ which satisfy the Clifford algebra, $\hat{\sigma}_i \hat{\sigma}_j + \hat{\sigma}_j \hat{\sigma}_j = 2\delta_{i,j}$. In particular, such algebra can be realized by the matrices

$$\hat{\sigma}_{2k-1} = \left(\otimes^{k-1}\hat{\sigma}_z\right) \otimes \hat{\sigma}_x \otimes \mathbb{1}_{2^{\lfloor d/2 \rfloor - k}}, \quad \hat{\sigma}_{2k} = \left(\otimes^{k-1}\hat{\sigma}_z\right) \otimes \hat{\sigma}_y \otimes \mathbb{1}_{2^{\lfloor d/2 \rfloor - k}}, \quad 1 \leq k \leq \lfloor d/2 \rfloor \tag{4}$$

and

$$d \text{ even}: \quad \hat{\sigma}_{d+1} = \otimes^{\lfloor d/2 \rfloor}\hat{\sigma}_z, \tag{5a}$$

$$d \text{ odd}: \quad \hat{\sigma}_d = \otimes^{\lfloor d/2 \rfloor}\hat{\sigma}_z. \tag{5b}$$

In the rest of the paper we will omit the tensor product $\otimes$ with those auxiliary degrees of freedom to simplifies the equations.

If $\hat{H}$ exhibits a local spectral gap $\Delta$ near $x'$, which requires that its gapless modes must be located far away from $x'$, we can show that $\hat{L}$ remains gapped for appropriately chosen $\kappa$. To show this, we square the localizer

$$d \text{ even}: \quad \hat{L}^2 = \hat{H}^2 + \kappa^2 \sum_{k=1}^{d}(\hat{x}_k - x_k')^2 + \sum_{k=1}^{d} \kappa[\hat{H}, \hat{x}_k]\hat{\sigma}_{d+1}\hat{\sigma}_k \tag{6a}$$

$$d \text{ odd}: \quad \hat{L}^2 = \hat{H}^2 + \kappa^2 \sum_{k=1}^{d}(\hat{x}_k - x_k')^2 + \sum_{k=1}^{d} \kappa[\hat{H}, \hat{x}_k]\hat{C}\hat{\sigma}_k. \tag{6b}$$

The first two terms are positive definite so their sum $\hat{Q} = \hat{H}^2 + \kappa^2 \sum_k (\hat{x}_k - x_k')^2$, known as the quadratic composite operator [103], is also positive definite. The remaining term, linear in $\kappa$, is bounded in norm by $\kappa \sum_k \|[\hat{H}, \hat{x}_k]\|$. The spectral localizer $\hat{L}$ therefore remains gapped provided $\hat{Q}$ has a gap larger than $\kappa \sum_k \|[\hat{H}, \hat{x}_k]\|$.

To determine the values of $\kappa$ for which this can happen, we can use the fact that the gap of $\hat{Q}$ cannot be much smaller than $\min(\Delta^2, \kappa^2 R^2)$ where $R$ is the distance from the gapless boundary mode to $x'$. This constraint follows from a contradiction argument: any hypothetical eigenmode $|\psi\rangle$ of $\hat{Q}$ with eigenvalue $\lambda$ much below $\min(\Delta^2, \kappa^2 R^2)$ would simultaneously need to verify $\langle\psi|\hat{H}^2|\psi\rangle \leq \lambda \ll \Delta^2$ and $\langle\psi|\kappa^2(\hat{x} - x')^2|\psi\rangle \leq \lambda \ll \kappa^2 R^2$. The former condition restricts the state to be inside the bulk gap, which by definition necessarily is a boundary state, while the latter requires the state to have negligible weight near the boundary, hence the contradiction. Therefore we know that $\hat{L}$ is gapped as long as $\min(\Delta^2, \kappa^2 R^2) \gg \kappa \sum_k \|[\hat{H}, \hat{x}_k]\|$ or equivalently

$$\frac{\sum_k \|[\hat{H}, \hat{x}_k]\|}{R^2} \ll \kappa \ll \frac{\Delta^2}{\sum_k \|[\hat{H}, \hat{x}_k]\|}, \tag{7}$$

which qualitatively recovers known rigorous results [103].

For simplicity in the following analysis, we will take $x'$ in the center of the bulk and set the origin such that $x' = 0$. We will then consider a large sample where the distance $R$ from the boundary to the center tends to infinity $R \to \infty$ so that the lower bound on $\kappa$ can be dismissed.

Since the spectral localizer has a gap around the zero energy, we can separate its eigenvalues into positive and negative eigenvalues. The spectral localizer index $I_{\text{SL}}$ is then defined as half the signature of $\hat{L}$: the number of positive eigenvalue minus the number of negative ones divided by two. Defining the flattened localizer operator $\hat{L}_{\text{F}} = \hat{L}(\hat{L}^2)^{-1/2}$, where all the positive and negatives eigenvalues of $\hat{L}$ are mapped to $+1$ and $-1$, respectively, the index can also be expressed as the trace of $\hat{L}_{\text{F}}$

$$I_{\text{SL}} = \frac{1}{2}\text{Sig}(\hat{L}) = \frac{1}{2}\text{Tr}'(\hat{L}_{\text{F}}) = \frac{1}{2}\text{Tr}'(\hat{L}(\hat{L}^2)^{-1/2}), \tag{8}$$

where the trace $\text{Tr}'$ is taken over the additional Clifford degrees of freedom in addition to the real-space trace. This index is the topological invariant for both classes A and AIII, provided we choose $\hat{L}$ according to dimensionality, as defined in Eq. (3).

## 3 Equivalence between the local Chern marker and the spectral localizer index

To map the spectral localizer index in Eqs. (8) to the Chern or winding markers in Eqs. (2), we Taylor expand $(\hat{L}^2)^{-1/2}$ in powers of $\kappa$. We use the fact that, when $\kappa$ is small, the third term in both Eqs. (6) is small compared to the gap imposed by the first two terms.

Without changing the spectral localizer, it is possible to replace $\hat{H}$ by a flattened Hamiltonian $\hat{H}_F = f(H)$. Here, $f$ is a smooth function that rescales the eigenstate energy $E$ above and below the gap to $f(E) = 1$ and $f(E) = -1$, respectively, smoothly interpolating between these values for in-gap gapless modes. In particular $\hat{H}_F$ verifies $\hat{H}_F^2 = \mathbb{1}$ in the bulk. These properties will simplify our computations.

For clarity, we present the detailed calculation for even dimensions in the main text while the almost identical odd-dimensional case is deferred to Appendix B. In the even-dimensional case, the Taylor expansion of $(\hat{L}^2)^{-1/2}$ in Eq. (8) takes the form

$$\begin{aligned}I_{\text{SL}} &= \frac{1}{2}\text{Tr}'\left(\hat{L}\left(\hat{H}_F^2 + \kappa^2\hat{r}^2 + \kappa\sum_{k=1}^{d}[\hat{H}_F, \hat{x}_k]\hat{\sigma}_{d+1}\hat{\sigma}_k\right)^{-1/2}\right) \\ &= \frac{1}{2}\text{Tr}'\left(\left(\hat{H}_F\hat{\sigma}_{d+1} + \kappa\sum_{k=1}^{d}\hat{x}_k\hat{\sigma}_k\right)\hat{g}^{1/2}\sum_n c_n\left(\hat{g}\kappa\sum_{k=1}^{d}[\hat{H}_F, \hat{x}_k]\hat{\sigma}_{d+1}\hat{\sigma}_k\right)^n\right),\end{aligned} \tag{9}$$

where $c_n = \frac{(-1)^n(2n)!}{4^n(n!)^2} = \frac{(-1)^n\Gamma(n+1/2)}{n!\sqrt{\pi}}$ and $\hat{g} = (\hat{H}_F^2 + \kappa^2\hat{r}^2)^{-1} = \hat{Q}^{-1}$ with $\hat{r}^2 = \sum_k \hat{x}_k^2$ and $\hat{Q}$ is the quadratic composite operator defined before which is therefore invertible.

When $r \gg 1/\kappa$, $\hat{g}$ becomes small, effectively restricting the real-space trace to a ball of radius $r \lesssim 1/\kappa$ of volume $\sim 1/\kappa^d$. So, the trace at most involves a sum over $O(1/\kappa^d)$ terms of order $\kappa^n$ giving a global scaling in $O(\kappa^{n-d})$. Therefore, in the $\kappa \to 0$ limit, the expansion can be truncated at $n = d$ since higher-order terms become negligible. Although many terms remain for $n \le d$, most of them vanish under the trace due to the Clifford algebra structure. Specifically, for any operator $A \otimes \prod_i \hat{\sigma}_{k_i}$, the trace is nonzero only if $\prod_i \hat{\sigma}_{k_i}$ is proportional to the identity, in which case the trace simplifies to $\text{Tr}'(A \otimes \mathbb{1}) = 2^{\lfloor d/2 \rfloor}\text{Tr}(A)$.

In the perturbative expansion, each product contains an odd number of $\sigma$ matrices. For any term to be nonzero every $\sigma$ matrix must appear at least once in the product. This is because

the only way that the product is equal to the identity matrix is by virtue of the mathematical identity $\prod_{i=1}^{d+1} \hat{\sigma}_i = i^{\lfloor d/2 \rfloor}$ . In particular, any simplification using the identity $\hat{\sigma}_i^2 = 1$ cannot change the parity of the number of Clifford matrices in the product. These observations imply that the only terms remaining after tracing over the Clifford degrees of freedom are

$$
\begin{aligned}
I_{\text{SL}} = &\frac{c_d}{2}\kappa^d(-2i)^{\lfloor d/2 \rfloor} \sum_{i_1,\dots,i_d} \epsilon_{\vec{i}} \text{Tr}\left( \hat{H}_F \hat{g}^{1/2} \prod_{k=1}^{d} \left(\hat{g}[\hat{H}_F,\hat{x}_{i_k}]\right) \right) \\
&- \frac{c_{d-1}}{2}\kappa^{d-1}(-2i)^{\lfloor d/2 \rfloor} \sum_{i_1,\dots,i_d} \epsilon_{\vec{i}} \text{Tr}\left( \kappa\hat{x}_{i_1}\hat{g}^{1/2} \prod_{k=2}^{d} \left(\hat{g}[\hat{H}_F,\hat{x}_{i_k}]\right) \right).
\end{aligned}
\tag{10}
$$

The term on the second line can be shown to vanish (see appendix A) so we focus on the first term. This term is proportional to the Chern marker as it can be rearranged into

$$
I_{\text{SL}} = \frac{c_d}{2}\kappa^d(-2i)^{\lfloor d/2 \rfloor} \sum_{i_1,\dots,i_d} \epsilon_{\vec{i}} \text{Tr}\left( \frac{1}{(1+\kappa^2\hat{r}^2)^{d+1/2}} \hat{H}_F \prod_{k=1}^{d} [\hat{H}_F,\hat{x}_{i_k}] \right),
\tag{11}
$$

where we first used that any commutator with $\hat{g} = (\hat{H}_F^2 + \kappa^2\hat{r}^2)^{-1}$ would create a higher order term in $\kappa$ that can be neglected. And secondly we used the identity $\hat{H}_F^2 = 1$ which is justified by the fact that $(\hat{H}_F^2 + \kappa^2\hat{r}^2)^{-(d+1/2)}$ decays fast enough when $r \gg 1/\kappa$ to make the trace convergent meaning the largest contribution to the trace comes from the bulk. Next, we can rewrite Eq. (11) in terms of the weight function

$$
w = \frac{(1+\kappa^2\hat{r}^2)^{-(d+\frac{1}{2})}}{\text{Tr}\left((1+\kappa^2\hat{r}^2)^{-(d+\frac{1}{2})}\right)},
\tag{12}
$$

which adds the overall prefactor

$$
\lim_{\kappa\to 0} \text{Tr}\left((1+\kappa^2\hat{r}^2)^{-(d+\frac{1}{2})}\right) = \int dx^d \frac{1}{(1+\kappa^2 r^2)^{d+\frac{1}{2}}} = \frac{\pi^{d/2}\Gamma(\frac{d+1}{2})}{\kappa^d \Gamma(d+\frac{1}{2})}.
\tag{13}
$$

Combining this coefficient with the rest of the prefactors, one can verify that the proportionality constant is one and we reach the desired equality between Eq. (11) and the averaged Chern marker Eq. (2a)

$$
I_{\text{SL}} = C_{d/2}.
\tag{14}
$$

Following similar steps for the odd dimensional case (see Appendix B) we obtain

$$
I_{\text{SL}} = W_{\lceil d/2 \rceil}.
\tag{15}
$$

Eqs. (14) and (15) are the main result of this work.

## 4  Conclusion and outlook

We have demonstrated a direct equivalence between the spectral localizer index $I_{\text{SL}}$ and the Chern and winding markers of topological classes A and AIII through a systematic perturbative expansion in the small-parameter regime, $\kappa \ll \frac{\Delta^2}{\sum_k \|[\hat{H},\hat{x}_k]\|}$, where $\Delta$ is the bulk gap of the Hamiltonian. By leveraging the Clifford algebra symmetries inherent to the spectral localizer's construction, we showed that for even dimensions, $I_{\text{SL}}$ reduces to the Chern marker $C_{d/2}$ as the leading-order contribution in $\kappa$, while in odd dimensions it analogously reduces the winding

marker $W_{\lceil d/2 \rceil}$. Our direct approach circumvents the need for abstract algebraic topology (e.g., K-theory or spectral flow), instead relying on transparent asymptotic analysis of the spectral localizer's resolvent. Our result provides an explicit proof of the equivalence between the spectral localizer and the Chern and winding marker. Additionally, this equivalence explains the numerically observed consistency between the topological phase diagrams calculated using the local Chern marker with those calculated with the spectral localizer index at small $\kappa$ [67].

Here we focused on $\mathbb{Z}$-classified topological phases via Chern and winding markers. A natural open question is to extend the methodology to include $\mathbb{Z}_2$ topological phases (e.g., time-reversal symmetric or particle-hole symmetric systems), to link the corresponding spectral localizer index [93] with existing local markers [75, 108–110].

## Acknowledgments

We thank A. Akhmerov, I. Araya Day, P. Delplace, P. d'Ornellas, C. Fulga, J. D. Hannukainen, and T. Loring for insightful discussions and related collaborations.

**Funding information**    A.G.G. acknowledges financial support from the European Research Council (ERC) Consolidator grant under grant agreement No. 101042707 (TOPOMORPH). L.J and J.B acknowledges financial support by the Swedish Research Council (VR) through Grant No. 2020-00214, and the European Research Council (ERC) under the European Union's Horizon 2020 research and innovation program (Grant Agreement No. 101001902).

## A    Vanishing of the second term of (10)

In this Appendix we show that the term on the second line of Eq. (10) vanishes. That term can be written as

$$
\begin{aligned}
B &= \frac{c_{d-1}}{2} \kappa^{d-1} (-2i)^{\lfloor d/2 \rfloor} \sum_{i_1,\ldots,i_d} \epsilon_{\vec{i}} \operatorname{Tr}\left( \kappa \hat{x}_{i_1} \hat{g}^{1/2} \prod_{k=2}^{d} \left( \hat{g}[\hat{H}_F, \hat{x}_{i_k}] \right) \right) \\
&= c \sum_{i_1,\ldots,i_d} \epsilon_{\vec{i}} \operatorname{Tr}\left( \hat{f}_{i_1} \prod_{k=2}^{d} \left( \hat{g}[\hat{H}_F, \hat{x}_{i_k}] \right) \right),
\end{aligned}
\tag{16}
$$

where $c = c_{d-1}\kappa^{d-1}(-2i)^{\lfloor d/2 \rfloor}/2$, $\hat{f}_{i_1} = \kappa \hat{x}_{i_1}\hat{g}^{1/2}$ with $\hat{g} = (\hat{H}_F^2 + \kappa^2 \hat{r}^2)^{-1}$.

To show that this term vanishes, we will use the fact that the trace contains $d-1$ products of $\hat{g}$ which decays sufficiently fast to make the integral convergent and therefore the contributions from the boundary negligible. Therefore, the largest contribution to the trace comes from the bulk were we can use the identity $\hat{H}_F^2 = 1$ and obtain that:

$$
B = \frac{1}{2} c \sum_{i_1,\ldots,i_d} \epsilon_{\vec{i}} \operatorname{Tr}\left( \hat{H}_F \left[ \hat{H}_F, \hat{f}_{i_1} \prod_{k=2}^{d} \left( \hat{g}[\hat{H}_F, \hat{x}_{i_k}] \right) \right]_+ \right),
\tag{17}
$$

where we used the notation $[A,B]_+ = AB + BA$ for the anti-commutator and the cyclic property of the trace. Using the fact that $\hat{H}_F$ anti-commutes with the commutator $[\hat{H}_F, A]$ as

$$
[\hat{H}_F, [\hat{H}_F, A]]_+ = [\hat{H}_F^2, A] = [\mathbb{1}, A] = 0,
\tag{18}
$$

we can show that this expression can be decomposed into:

$$
\begin{aligned}
B = &\frac{c}{2} \sum_{i_1, \ldots, i_d} \epsilon_{\vec{i}} \operatorname{Tr} \left( \hat{H}_F \left[ \hat{H}_F, \hat{f}_{i_1} \right] \prod_{k=2}^{d} \left( \hat{g} [\hat{H}_F, \hat{x}_{i_k}] \right) \right) \\
&+ \frac{c}{2} \sum_{j=2}^{d} \sum_{i_1, \ldots, i_d} \epsilon_{\vec{i}} (-1)^j \operatorname{Tr} \left( \hat{H}_F \hat{f}_{i_1} \prod_{k=2}^{j-1} \left( \hat{g}[\hat{H}_F, \hat{x}_{i_k}] \right) \left( [\hat{H}_F, \hat{g}][\hat{H}_F, \hat{x}_{i_j}] \right) \prod_{k=j+1}^{d} \left( \hat{g}[\hat{H}_F, \hat{x}_{i_k}] \right) \right).
\end{aligned}
\tag{19}
$$

We can now simplify this expression by keeping only the leading order term in $\kappa$. In particular, for any function $F$, with corresponding diagonal operator $\hat{F}$, that is slowly varying in the limit $\kappa \to 0$, we have that

$$
\lim_{\kappa \to 0} \left[ \hat{H}_F, \hat{F} \right] = \sum_i (\partial_{x_i} \hat{F})[\hat{H}_F, \hat{x}_i].
\tag{20}
$$

In our case we can keep only the term $(\partial_{x_{i_1}} \hat{F})[\hat{H}_F, \hat{x}_{i_1}]$. The reason is that choosing $j \neq i_1$ leads to expressions where some commutator $[\hat{H}, \hat{x}_j]$ appears twice in the product. Using the cyclic property of the trace and anti-commutation of $[\hat{H}, \hat{x}_j]$ with $\hat{H}$, one can show that these contributions for $j \neq i_1$ vanish. Thus, keeping only $j = i_1$ terms we write

$$
\begin{aligned}
B = &\frac{c}{2} \sum_{i_1, \ldots, i_d} \epsilon_{\vec{i}} \operatorname{Tr} \left( \partial_{i_1} \hat{f}_{i_1} \hat{g}^{d-1} \hat{H}_F \prod_{k=1}^{d} [\hat{H}_F, \hat{x}_{i_k}] \right) \\
&+ \frac{c}{2} \sum_{j=2}^{d} \sum_{i_1, \ldots, i_d} \epsilon_{\vec{i}} (-1)^j \operatorname{Tr} \left( \hat{f}_{i_1} (\partial_{x_{i_1}} \hat{g}) \hat{g}^{d-2} \hat{H}_F \prod_{k=2}^{j-1} [\hat{H}_F, \hat{x}_{i_k}] [\hat{H}_F, \hat{x}_{i_1}] \prod_{k=j}^{d-1} [\hat{H}_F, \hat{x}_{i_k}] \right),
\end{aligned}
\tag{21}
$$

where we positioned the operators $\hat{f}_{i_1}$ and $\hat{g}$ on the left by neglecting the resulting commutators as they would be of higher-order in $\kappa$. Reordering the commutators using the cyclic property of the trace, we reduce this further to

$$
\begin{aligned}
B &= \frac{c}{2} \sum_{i_1, \ldots, i_d} \epsilon_{\vec{i}} \operatorname{Tr} \left( \partial_{x_{i_1}} \left( \hat{f}_{i_1} \hat{g}^{d-1} \right) \hat{H}_F \prod_{k=1}^{d} [\hat{H}_F, \hat{x}_{i_k}] \right) \\
&= \frac{c}{2} \sum_{i_1, \ldots, i_d} \epsilon_{\vec{i}} \operatorname{Tr} \left( \frac{1}{d} \sum_j \partial_{x_j} \left( \hat{f}_j \hat{g}^{d-1} \right) \hat{H}_F \prod_{k=1}^{d} [\hat{H}_F, \hat{x}_{i_k}] \right),
\end{aligned}
\tag{22}
$$

which is a term proportional to the Chern marker but with a vanishing proportionality constant since

$$
\lim_{\kappa \to 0} \operatorname{Tr} \left( \sum_j \partial_{x_j} \left( \hat{f}_j \hat{g}^{d-1} \right) \right) = \int dx^d \sum_{j=1} \partial_{x_j} \left( f_j g^{d-1} \right) = 0
\tag{23}
$$

because

$$
\hat{f}_j \hat{g}^{d-1} = \frac{\kappa x_j}{(1 + \kappa^2 r^2)^{d-1/2}} \to 0
\tag{24}
$$

in the limit $r \to +\infty$. Note that $d \geq 2$ since we are constrained to the topologically nontrivial cases in class A. This proves the wanted result that the second term of equation (10) vanishes.

## B  Chiral case

To arrive to Eq. (15) we write the perturbative expansion of

$$
\begin{aligned}
I_{\text{SL}} &= \frac{1}{2} \operatorname{Tr}\left( \hat{L}\left( \hat{H}_F^2 + \kappa^2 \hat{r}^2 + \kappa \sum_{k=1}^d [\hat{H}_F, \hat{x}_k]\hat{C}\hat{\sigma}_k \right)^{-1/2} \right) \\
&= \frac{1}{2} \operatorname{Tr}\left( \left( \hat{H}_F + \kappa \sum_{k=1}^d \hat{x}_k \hat{C}\hat{\sigma}_k \right) \hat{g}^{1/2} \sum_n c_n \left( \hat{g}\kappa \sum_{k=1}^d [\hat{H}_F, \hat{x}_k]\hat{C}\hat{\sigma}_k \right)^n \right)
\end{aligned}
\tag{25}
$$

As for the nonchiral case, the perturbative expansion can also be truncated at order $n = d$, see the discussion below Eq. (9). Similarly, for any operator $A \otimes \prod_i \hat{\sigma}_{k_i}$, the trace is nonzero only if $\prod_i \hat{\sigma}_{k_i}$ is proportional to the identity, where the trace simplifies to $\operatorname{Tr}'(A \otimes \mathbb{1}) = 2^{\lfloor d/2 \rfloor} \operatorname{Tr}(A)$.

The chiral symmetry condition $\hat{H}_F \hat{C} + \hat{C}\hat{H}_F = 0$ imposes strong constraints on the trace structure. Since $\hat{H}_F$ anticommutes with $\hat{C}$, any term in the expansion containing an odd number of $\hat{H}_F$ factors must vanish under the trace operation. This immediately eliminates half of the potential contributing terms. For the remaining terms with even products of $\hat{H}_F$, careful examination reveals that they necessarily contain an odd number of $\sigma_i$ matrices. So, as in the main text, the only way that the product of an odd number of Clifford matrices is proportional to the identity is by using the identity $\prod_{i=1}^d \sigma_i = i^{\lfloor d/2 \rfloor}$. Hence, every matrix $\sigma$ must appear at least once in the product. After tracing out the Clifford degrees of freedom, only the following terms remain

$$
\begin{aligned}
I_{\text{SL}} =&\frac{c_d}{2}\kappa^d(-2i)^{\lfloor d/2 \rfloor} \sum_{i_1,\dots,i_d} \epsilon_{\vec{i}} \operatorname{Tr}\left( \hat{C}\hat{H}_F \hat{g}^{1/2} \prod_{k=1}^d \left( \hat{g}[\hat{H}_F, \hat{x}_{i_k}] \right) \right) \\
&- \frac{c_{d-1}}{2}\kappa^{d-1}(-2i)^{\lfloor d/2 \rfloor} \sum_{i_1,\dots,i_d} \epsilon_{\vec{i}} \operatorname{Tr}\left( \hat{C}\kappa\hat{x}_{i_1} \hat{g}^{1/2} \prod_{k=2}^d \left( \hat{g}[\hat{H}_F, \hat{x}_{i_k}] \right) \right).
\end{aligned}
\tag{26}
$$

We show below that the term on the second line vanishes, so only the first term remains. This term is proportional to the Chern marker as it can be rearranged into

$$
I_{\text{SL}} = \frac{c_d}{2}\kappa^d(-2i)^{\lfloor d/2 \rfloor} \sum_{i_1,\dots,i_d} \epsilon_{\vec{i}} \operatorname{Tr}\left( \frac{1}{(\mathbb{1} + \kappa^2 \hat{r}^2)^{d+1/2}} \hat{C}\hat{H}_F \prod_{k=1}^d [\hat{H}_F, \hat{x}_{i_k}] \right),
\tag{27}
$$

where we first used that any commutator with $\hat{g} = (\hat{H}_F^2 + \kappa^2 \hat{r}^2)^{-1}$ would create a higher order term in $\kappa$ that can be neglected. And secondly we used the identity $\hat{H}_F^2 = \mathbb{1}$ which is justified by the fact that $(\hat{H}_F^2 + \kappa^2 \hat{r}^2)^{-(d+1/2)}$ decays fast enough when $r \gg 1/\kappa$ to make the trace convergent meaning the largest contribution to the trace comes from the bulk. Eq. (11) is a winding marker (2a) for a weight function $w = (\mathbb{1} + \kappa^2 \hat{r}^2)^{-(d+\frac{1}{2})}/\operatorname{Tr}\left( (\mathbb{1} + \kappa^2 \hat{r}^2)^{-(d+\frac{1}{2})} \right)$ where we have that

$$
\lim_{\kappa \to 0} \operatorname{Tr}\left( (\mathbb{1} + \kappa^2 \hat{r}^2)^{-(d+\frac{1}{2})} \right) = \int dx^d \frac{1}{(1 + \kappa^2 r^2)^{d+\frac{1}{2}}} = \frac{\pi^{d/2}\Gamma(\frac{d+1}{2})}{\kappa^d \Gamma(d + \frac{1}{2})}.
\tag{28}
$$

Combining this coefficient with the other prefactors, one verifies that the proportionality constant is one and we have the equality

$$
I_{\text{SL}} = W_{\lceil d/2 \rceil},
\tag{29}
$$

which is Eq. (15).

We now show that the second term in Eq. (26) vanishes. This term can be written as

$$
B = \frac{c_{d-1}}{2} \kappa^{d-1} (-2i)^{\lfloor d/2 \rfloor} \sum_{i_1,\dots,i_d} \epsilon_{\vec{i}} \, \mathrm{Tr}\left( \hat{C} \kappa \hat{x}_{i_1} \hat{g}^{1/2} \prod_{k=2}^{d} \left( \hat{g}[\hat{H}_F, \hat{x}_{i_k}] \right) \right)
$$

$$
= c \sum_{i_1,\dots,i_d} \epsilon_{\vec{i}} \, \mathrm{Tr}\left( \hat{C} \hat{f}_{i_1} \prod_{k=2}^{d} \left( \hat{g}[\hat{H}_F, \hat{x}_{i_k}] \right) \right),
$$

(30)

where $c = c_{d-1} \kappa^{d-1} (-2i)^{\lfloor d/2 \rfloor}/2$ and $\hat{f}_{i_1} = \kappa \hat{x}_{i_1} \hat{g}^{1/2}$. To show that the term vanishes, we use the fact that the trace contains $d-1$ products of $\hat{g}$ which make the trace integrable except when $d = 1$. Leaving the case $d = 1$ aside for now, for $d > 1$ the trace is convergent so most of the contribution comes from the bulk where we can insert the identity $\hat{H}_F^2 = \mathbb{1}$, and obtain that

$$
B = \frac{c}{2} \sum_{i_1,\dots,i_d} \epsilon_{\vec{i}} \, \mathrm{Tr}\left( \hat{C} \hat{H}_F \left[ \hat{H}_F, \hat{f}_{i_1} \prod_{k=2}^{d} \left( \hat{g}[\hat{H}_F, \hat{x}_{i_k}] \right) \right] \right),
$$

(31)

where we used the notation $[A, B]_+ = AB + BA$ for the anti-commutator.

Using the fact that $\hat{H}_F$ anti-commutes with the commutator $[\hat{H}_F, A]$ as

$$
[\hat{H}_F, [\hat{H}_F, A]]_+ = [\hat{H}_F^2, A] = [\mathbb{1}, A] = 0,
$$

(32)

we decompose this expression into

$$
B = \frac{c}{2} \sum_{i_1,\dots,i_d} \epsilon_{\vec{i}} \, \mathrm{Tr}\left( \hat{C} \hat{H}_F [\hat{H}_F, \hat{f}_{i_1}] \prod_{k=2}^{d} \left( \hat{g}[\hat{H}_F, \hat{x}_{i_k}] \right) \right)
$$

$$
+ \frac{c}{2} \sum_{j=2}^{d} \sum_{i_1,\dots,i_d} \epsilon_{\vec{i}} (-1)^j \, \mathrm{Tr}\left( \hat{C} \hat{H}_F \hat{f}_{i_1} \prod_{k=2}^{j-1} \left( \hat{g}[\hat{H}_F, \hat{x}_{i_k}] \right) \left( [\hat{H}_F, \hat{g}][\hat{H}_F, \hat{x}_{i_j}] \right) \prod_{k=j+1}^{d} \left( \hat{g}[\hat{H}_F, \hat{x}_{i_k}] \right) \right).
$$

(33)

We now simplify by keeping only terms to leading order in $\kappa$ using that for slowly varying function $F$, $\lim_{\kappa \to 0} [\hat{H}_F, \hat{F}] = \sum_i (\partial_{x_i} \hat{F})[\hat{H}_F, \hat{x}_i]$. In our case we can keep only the term $(\partial_{x_{i_1}} \hat{F})[\hat{H}_F, \hat{x}_{i_1}]$. The reason is that choosing $j \neq i_1$ leads to expressions where some commutator $[\hat{H}, \hat{x}_j]$ appears twice in the product. Using the cyclic property of the trace and the anti-commutation of $[\hat{H}, \hat{x}_j]$ with $\hat{H}$, one can show that the contributions for $j \neq i_1$ vanish. Keeping only the $j = i_1$ terms, this results in

$$
B = \frac{c}{2} \sum_{i_1,\dots,i_d} \epsilon_{\vec{i}} \, \mathrm{Tr}\left( \hat{C} \partial_{i_1} \hat{f}_{i_1} \hat{g}^{d-1} \hat{H}_F \prod_{k=1}^{d} [\hat{H}_F, \hat{x}_{i_k}] \right)
$$

$$
+ \frac{c}{2} \sum_{j=2}^{d} \sum_{i_1,\dots,i_d} \epsilon_{\vec{i}} (-1)^j \, \mathrm{Tr}\left( \hat{C} \hat{f}_{i_1} (\partial_{x_{i_1}} \hat{g}) \hat{g}^{d-2} \hat{H}_F \prod_{k=2}^{j-1} [\hat{H}_F, \hat{x}_{i_k}] [\hat{H}_F, \hat{x}_{i_1}] \prod_{k=j}^{d-1} [\hat{H}_F, \hat{x}_{i_k}] \right),
$$

(34)

where we positioned the operators $\hat{f}_{i_1}$ and $\hat{g}$ on the left by neglecting the resulting commutators as they would be of higher order. Reordering the commutators using the cyclic property of the trace, this can be further reduced to

$$
B = \frac{c}{2} \sum_{i_1,\dots,i_d} \epsilon_{\vec{i}} \, \mathrm{Tr}\left( \hat{C} \partial_{x_{i_1}} \left( \hat{f}_{i_1} \hat{g}^{d-1} \right) \hat{H}_F \prod_{k=1}^{d} [\hat{H}_F, \hat{x}_{i_k}] \right)
$$

$$
= \frac{c}{2} \sum_{i_1,\dots,i_d} \epsilon_{\vec{i}} \, \mathrm{Tr}\left( \hat{C} \frac{1}{d} \sum_{j} \partial_{x_j} \left( \hat{f}_j \hat{g}^{d-1} \right) \hat{H}_F \prod_{k=1}^{d} [\hat{H}_F, \hat{x}_{i_k}] \right),
$$

(35)

which is a term proportional to the Chern marker but with a vanishing proportionality constant as

$$\text{Tr}\left(\sum_j \partial_{x_j}\left(\hat{f}_j \hat{g}^{d-1}\right)\right) \overset{\kappa \to 0}{=\!=\!=} \int dx^d \sum_{j=1} \partial_{x_j}\left(f_j g^{d-1}\right) = 0 \tag{36}$$

since

$$\hat{f}_j \hat{g}^{d-1} = \frac{\kappa x_j}{(1+\kappa^2 r^2)^{d-1/2}} \to 0 \tag{37}$$

in the infinite limit $r \to +\infty$ as long as we have $d > 1$. This therefore proves the wanted result that the second term of equation (26) vanishes for $d > 1$.

We finish by discussing the case $d = 1$. Starting from the initial expression of $B$ (30) in $d = 1$

$$B = c \,\text{Tr}\left(\hat{C}\hat{x}/(1+\kappa^2\hat{x}^2)\right) \tag{38}$$

we see that $B = 0$ as long as we have equally many internal degrees of freedom of positive and negative chirality, which is a common case studied in the literature. However if there is a chirality imbalance between the internal degrees of freedom of opposite chirality, $B$ will not vanish and will correct the value of the marker. This is a known phenomena particular to one-dimensional topological insulators, see e.g. [44, 79, 111–113].

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
