# Peer review of "Explicit equivalence between the spectral localizer and local Chern and winding markers"

_SciPost Physics_

## Round 1 · Referee Report · Vladimir A. Zakharov (Referee 2) · 2025-11-10

The referee discloses that the following generative AI tools have been used in the preparation of this report:
Grammar check
Report
In (3b), the authors present a formula for the spectral localizer in odd dimensions, referring to [83, 103] (83. T.A. Loring, K-theory and pseudospectra for topological insulators, Annals of Physics (2015); 103. A. Cerjan, T.A. Loring, Classifying photonic topology using the spectral localizer and numerical K-theory, APL Photonics (2024)). The authors’ definition does not agree with the one given in Eq. (19) of [103]. Specifically, in (3b) the Hamiltonian matrix is multiplied by an identity matrix, which cannot be used to form a Clifford representation. In [103], the size of the Clifford matrices is required to be 2^{⌈d/2⌉} to guarantee irreducibility, while in (3b) it is 2^{⌊d/2⌋}. Additionally, in (3b) the position matrices are multiplied by a matrix C representing chiral symmetry, which is not present in the general form of the spectral localizer.
Moreover, as discussed in Sec. IIID of [103], the signature of the spectral localizer in odd dimensions is always zero and thus cannot serve as a topological marker. This suggests that in (3b) the authors intended to define the symmetry-reduced spectral localizer (28) of [103], which can be used to detect a local winding number (27) [103] (see also Sec. 4.1 [83]). However, the form used by the authors differs from that in the references: they multiply only the position matrices by the chiral matrix on the right, whereas in Loring’s definitions both the Hamiltonian and position matrices are multiplied by it.
Lastly, references [83] and [103] provide explicit formulas for a local topological marker for the AIII class obtained from the reduced spectral localizer only in 1D, with no general formula given for higher dimensions.
Therefore, I believe a minor revision is required to (i) clearly define the symmetry-reduced spectral localizer, (ii) prove or discuss (or cite appropriate references for) its equivalence to Loring’s form, and (iii) justify its use in dimensions higher than 1.
P.S. Typo: In the introduction, in the third paragraph, the letter T is missing at the beginning of the second sentence.
Recommendation
Ask for minor revision
Report
This manuscript presents a derivation of the equivalence between the local topological markers found using the spectral localizer framework and those local markers found using prior constructions. While this equivalence had been known previously, the presentation here is provided in a much more physically intuitive manner and thus I generally recommend this manuscript for publication. Indeed, if this journal has any sort of "editor's suggestion" demarcation, I would recommend this manuscript for that distinction.
Prior to publication, I would encourage the authors to consider and comment on a few points.
1) The end result of the calculation, Eqs. 14 and 15 strike me as slightly odd. My general expectation of working with traditional local Chern markers is that these are, in practice, not integer valued, as the weighting function in Eqs. 2a,b needs to somewhat fine-tuned to find quantized results on a computer. Adding disorder further complicates this, as it may really introduce domains with different local topology. Altogether, I would encourage the authors to clarify these points.
2) The authors are achieving a great deal of utility in recasting the spectral localizer to use the flattened Hamiltonian and localizer, which is awesome. But, it might be illuminating to see what this sequence of approximations are doing to the spectrum of L. E.g., if I take the "guts" of Eq. 9, 10, and 11, and shove everything (except a 1/2) back into the Trace, I could then look at the spectrum of the approximated flattened localizer, and see how this changes as the various approximations are implemented.
I would strongly encourage the authors to model a simple system and provide a series of plots of the spectrum of L, L_F, etc.
3) The manuscript has quite a few typos. An incomplete list: - Sec 1, paragraph 3, sentence 2, "o" -> "To". - missing space after comma in the line after Eq. 1b. - "such an algebra" line before Eq. 4.
Recommendation
Ask for minor revision

---

## Editorial Decision

unknown